# Musculoskeletal pain affects the age of retirement and the risk of work cessation among older people

**Nils Georg Niederstrasser**[1]*, **Elaine Wainwright**[2], **Martin J. Stevens**[2]

**1** Department of Psychology, University of Portsmouth, Portsmouth, United Kingdom, **2** Aberdeen Centre for Arthritis and Musculoskeletal Health (Epidemiology Group), University of Aberdeen, Aberdeen, United Kingdom

* nils.niederstrasser@port.ac.uk

**Data Availability Statement:** The ELSA dataset is freely available from the UK Data Service to all bonafide researchers. The dataset can be accessed here: https://discover.ukdataservice.ac.uk/series/?sn=200011.

## Abstract

### Objectives

Many people with chronic pain cannot work, while working despite chronic pain is linked to absenteeism and presenteeism and a host of other deleterious effects. This disproportionately affects older adults, who are closer to retirement, while the exact relationship between pain and work cessation as well as retirement among older adults is not known. We explore longitudinally the relationship between chronic pain and the risk of ceasing work and entering retirement.

### Methods

Data from 1156 individuals 50 years or older living in England taking part in the English Longitudinal Study of Ageing were used in this study. Cox proportional hazards regression analyses were used to examine the nature of the relationship between musculoskeletal pain and work cessation as well as retirement longitudinally over the course of fourteen years.

### Results

Suffering from frequent musculoskeletal pain was associated with an increased risk of ceasing work and retiring at an earlier age, as did work dissatisfaction, higher perceived social status, female gender, and not receiving the recognition they felt they deserved in their job. Severity of depressive symptoms, psychosocial job demands, decision authority, and social support did not influence the age at which participants reported work cessation or retirement.

### Conclusions

Frequent musculoskeletal pain may increase the risk of earlier work exit and earlier retirement. Further research should establish the mechanisms and decision making involved in leaving the workforce in people with frequent musculoskeletal pain.

**Funding:** The UK Data Archive made available the data. A team of researchers based at University College London, NatCen Social Research, the Institute for Fiscal Studies and the University of Manchester developed the English Longitudinal Study of Ageing.NatCen Social Research collected the data. The National Institute of Aging (R01AG017644) and a consortium of UK government departments coordinated by the Economic and Social Research Council provide funding for ELSA. ELSA is funded by the National Institute on Aging (R01AG017644), and by UK Government Departments coordinated by the National Institute for Health and Care Research (NIHR). The funders had no role in the study design; in the collection, analysis, and interpretation of data; in writing of the report; or in the decision to submit the paper for publication. The developers and funders of ELSA and the Archive do not bear any responsibility for the analyses or interpretations presented here.

**Competing interests:** The authors have declared that no competing interests exist.

## Introduction

Chronic pain is defined as pain that may fluctuate but lasts for three months or more [1]. Chronic pain, in particular musculoskeletal (MSK) pain such as low back pain, is a leading cause of disability worldwide [2]. Work which is physically and psychologically safe is good for our health and wellbeing [3,4]. This is still the case for people who live with chronic pain, including chronic MSK pain [3,5,6]. Yet up to 42% of people with chronic pain cannot work [7] and people who live with chronic pain are significantly more likely to leave work than people without pain [8]. People who are not in employment because of ill health or disability are more likely to have chronic pain than those who are employed [9]. Chronic pain impacts on those who live with it who remain at work, as it is linked to absenteeism and presenteeism [10,11]. It is also associated with a range of deleterious impacts for the worker including reduced working capacity [12], reduced income [13], and experiencing stigma for being seen as an unproductive worker [10,14].

There is an added important dimension when we consider older workers with pain. Since the prevalence of people living with chronic MSK pain conditions increases as we age, pain-related disability is a problem which is growing concomitant with aging populations in many countries including the UK [15]. Older workers may be defined as those aged 50 or over [16] as this represents the age at which labour force participation starts to reduce [17] although there is variation in age cuts off e.g., Leijten et al [18] used people aged 45–64. Older workers who have recently moved into economic inactivity due to the Covid pandemic were more likely to find themselves in poverty [19]. This combined with the longstanding agenda to extend working lives [20] has thrown into sharp relief the need to continue working for many people, including those in pain.

There are many cross-sectional studies that examine associations between living with chronic pain and employment status and outcomes [9]. However, there are very few longitudinal analyses of older workers with pain. An exception is by Leijten et al. [18]. They assessed the influence of seven chronic health problems, one of which was musculoskeletal pain, on work ability and productivity at work, via longitudinal analyses with data from older workers aged 45–64. They found that workers with musculoskeletal and mental health issues had lower productivity at work at one year follow up than workers without those health problems. Such analyses can be important for understanding the trajectories people living with pain experiences in their working lives, in terms of impact of pain on work, work retention, and retirement decisions. Longer time periods may be helpful to understand with more granularity the relationship between chronic pain and its interference with work as well as the risk of work cessation and retirement decisions, all key aspects of issues with which the older citizen with chronic pain must currently contend.

Since 2011 there are no general provisions for a mandatory retirement age in the UK, meaning that most people are at least notionally able to choose when to retire [21]. This is coupled with changes to the age at which people can receive state pension benefits which has been increasing incrementally since 2010 towards a current target of 68. However, despite the notional choice it is clear that people in less favourable work scenarios may choose to leave work earlier and/or be pushed out of their work. The decision to retire is multi-factorial, which may include health considerations and work factors [22,23].

Therefore, we explored longitudinally the relationship between chronic pain and the risk of work cessation; between chronic pain and how old people are when they retire.

Precise research questions were: does living with musculoskeletal pain affect the age of retirement and the risk of work cessation, while controlling for the influence of job satisfaction, depressive symptoms, self-perceived social status, sex, wealth, and working conditions.

## Methods

### Study design and setting

This longitudinal cohort study draws on data from participants taking part in the English Longitudinal Study of Ageing (ELSA), a national panel study comprising participants aged 50 years and older residing in England. Participants are followed over successive waves (2-year intervals) and, in this study, we draw on data from ELSA waves 2 (2004/2005) through 9 (2018/2019).

### Participants

Participants taking part in ELSA were initially taken from the Health Survey for England (HSE) for wave 1. Samples from the HSE stemming from 1998, 1999, and 2001 were used to recruit participants for ELSA, as these were deemed recent and large enough samples. Only those households where there was at least one adult over 50 years of age who had consented to being re-contacted subsequent to taking part in the HSE were approached for inclusion in the ELSA study. The ELSA sample has been designed specifically to represent individuals residing in private households aged 50 or older in England, as such participants were only included if they fulfilled these criteria. Given the respondents' age, ELSA data need to be refreshed each certain waves with new participants taken from the HSE to maintain its representative nature. This occurred at waves 3, 4, 6, 7, and 9. Additional details relating to sampling are described in Steptoe et al. [24].

Participants were eligible for inclusion in this study if they had complete data for all relevant variables spanning waves 2 to 9 and had not yet retired or were in work at wave 2. The entire data set at baseline (wave 2) comprised 8780 participants out of which 2405 reported being in employment and 2582 reported not being retired and so were eligible for inclusion in the study. After removal of missing values due to incomplete data and loss to follow up the final numbers were 1281 and 1156 eligible participants respectively. Out of the eligible participants, 1156 belonged to both categories, i.e., currently in employment and not yet retired (Fig 1). Full written consent was obtained from participants prior to taking part, while the London Multi-centre Research Ethics Committee (MREC/01/2/91) granted ethical approval for the data collection and archiving.

### Variables

In ELSA, participants either self-report data and/or data are collected through nurse visits. Overall, ELSA aims to collect data about the same topics at each wave, but there have been additions and omissions regarding the topics over successive waves. All visits are conducted face-to-face, whereby the data collection process is facilitated through the use of computer assisted interviewing overseen by a qualified nurse or using pen and paper for self-completion questionnaires. Interviews are conducted individually or in case of multiple eligible individuals in a single household concurrently in a single session. Additional details relating to data collection methods are described in Steptoe et al. [24].

### Outcomes

**Work cessation.** To determine whether participants had been employed in the last month they were asked to indicate whether they were in paid employment last month (yes, no).

**Retirement.** Participants self-reported whether they were retired. They were asked to select from a list what best describes their situation, i.e., retired, employed, self-employed, unemployed, permanently sick or disabled, looking after home or family, other, and semi-retired. We considered participants to be retired if they had selected "retired" or "semi-retired"

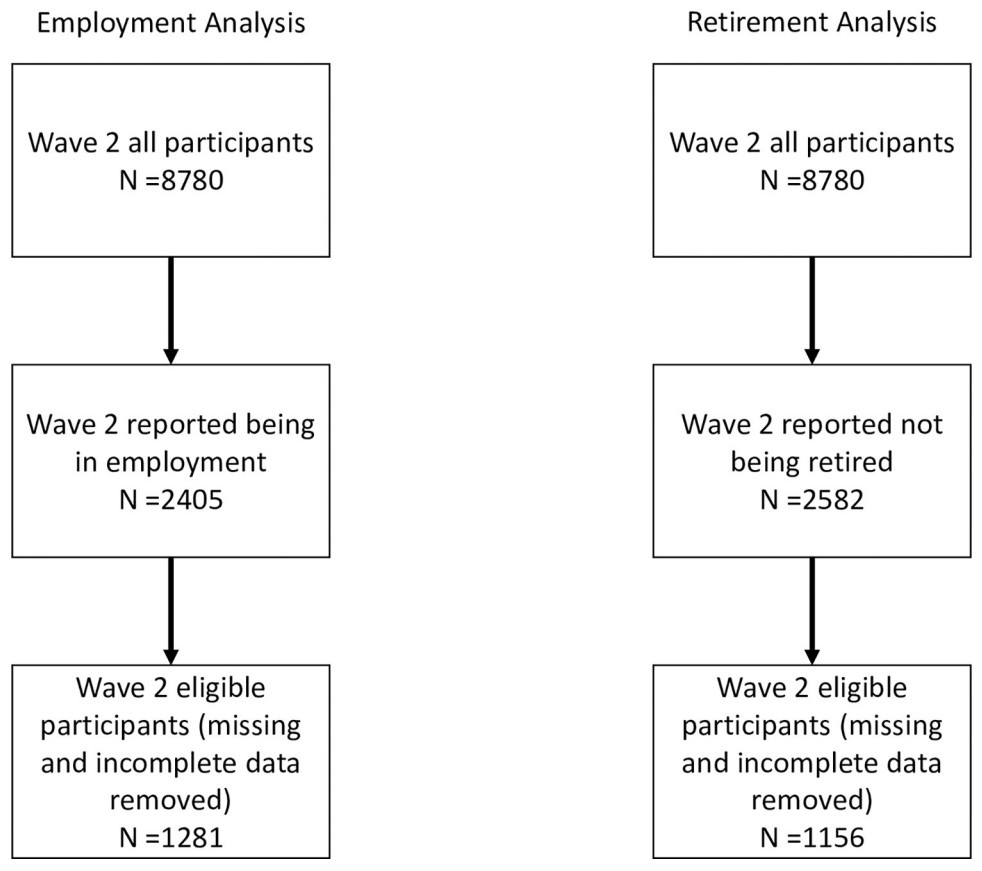

**Fig 1. Participant flow diagram.**

from the list as best describing their current situation. All other responses were coded as not retired.

## Predictors

**Job satisfaction.** Participants indicated whether they felt satisfied with their job, selecting from strongly agree, agree, disagree, and strongly disagree. To facilitate analysis and interpretation, the item was dichotomised, coding "strongly agree", and "agree" as a positive response and strongly disagree and disagree as a negative response.

**Depressive symptoms.** Participants completed the Centre for Epidemiologic Studies Depression Scale (CES-D), an eight-item measure assessing depressive symptoms [25]. Participants report the presence of absence of somatic complaints and negative affect they experienced in the previous week. The summed total score was divided by 8, resulting in scores ranging from 0 (no symptoms) to 1 (all symptoms).

**Self-perceived social status.** Participants were asked to rate their self-perceived (subjective) social status on a scale from 0 to 100, subdivided into 20 five-point increments. 0 represents those being "worst off" and 100 represents those being "best off". Scores range from 0 to 20.

**Frequent musculoskeletal pain.** Following previous investigations using the same data set [26,27], we used participants' self-reports (yes/no) to indicate whether they were often troubled by bone, joint, or muscle pain.

**Age.** Age was self-reported; however, if participants were older than 90 their ages were changed to 99 to maintain anonymity.

**Sex.** Participants self-reported their sex during nurse interviews.

**Wealth quintile.** First, participants' total net wealth was determined. This was the total of their housing wealth, physical wealth (including additional property wealth, wealth related to business and other physical assets), and financial wealth (including savings, stock certificates and bank accounts) minus any financial and mortgage debt. Then, based on the total net wealth, participants were organised into quintiles.

**Working conditions.** Based on the methodology laid out in Carr et al. [28] we calculated indexes of physical job demands, psychosocial job demands, decision authority, social support, and recognition. Physical job demands were derived by asking participants to rate their level of agreement with the statement "my job is physically demanding" ('strongly disagree', 'disagree', 'agree' or 'strongly agree') and the level of physical exertion in their current job, from sedentary (sitting) to heavy manual, resulting in sum scores ranging from 2–8 with higher scores indicating higher demand. The same approach was used for psychosocial demands. The extent of agreement with the items ("considering the things I have to do at work, I have to work very fast") and time pressure ("I am under constant time pressure due to a heavy workload") were summed (range 2–8). Decision authority was operationalised by summing the rate of agreement with the items on job control ("I feel I have control over what happens in most situations") and job autonomy ("I have very little freedom to decide how I do my work"; reversed) resulting in scores ranging from 2–8. Social support was measured through a binary item ("I receive adequate support in difficult situations"), as was low recognition ("I receive the recognition I deserve for my work"). For these items, responses of 'agree' or 'strongly agree' were coded as 0 and 'disagree' or 'strongly disagree' were coded as 1.

**Marital status.** Respondents indicated whether they were married, cohabiting, or neither. "Married" and "cohabiting" were coded into one category, whereas "neither" was used as the reference category.

## Statistical methods

R version 4.2.2 was used to run the statistical analyses. We performed Cox proportional hazards regressions to explore the relationship between the predictors and retirement as well as work cessation. Participants already retired or out of work at baseline (wave 2) were excluded from the respective analyses.

A single data set was derived, combining data from ELSA waves 2 through 9, whereby data from wave 2 (baseline) were used for the predictors. Participants' age was used to denote survival time until retiring or reporting work cessation. Data analysis in R was operationalised through the "survival" package and the results visualised using the "survminer" package [29]. Prior to running the multivariate model, individual univariate proportional hazard regressions for each potential predictor were run. Only those predictors significant in the univariate analysis were added to the multivariate model. Finally, additional sensitivity analyses were carried out to rule out reverse causation. This involved repeating the multivariate analyses while excluding participants who had retired/were out of work at baseline (wave 2) and those who retired or reported being out of work in wave 3.

## Results

### Sample characteristics

Table 1 presents an overview of the continuous variables employed in both analyses. The counts for the categorical variables can be found in Figs 2 and 3. Variations in sample size are due to differences in the number of complete responses.

**Table 1. Sample overview for measures at baseline.**

| Variables | Retirement Analysis (n = 1156) | Employment Analysis (n = 1281) |
|---|---|---|
| Age—mean (*SD; range*) | 57.8 (*3.9*); *52–75* | 58.3 (*4.5*); *52–99* |
| Self-described social status—mean (*SD*), range 0–20 | 12.4 (*2.9*) | 12.5 (*3.0*) |
| Psychosocial Job Demands—mean (*SD*), range 2–8 | 4.9 (*1.5*) | 5.0 (*1.5*) |
| Decision Authority—mean (*SD*), range 0–8 | 4.1 (*1.2*) | 4.1 (*1.2*) |
| Depressive Symptoms—mean (*SD*), range 0–1 | 0.1 (*0.2*) | 0.1 (*0.2*) |
| Recognition | Yes = 842 | Yes = 954 |
| | No = 314 | No = 327 |
| Social Support | Yes = 878 | Yes = 983 |
| | No = 278 | No = 298 |
| Sex | Male = 587 (50.8%) | Male = 637 (49.7%) |
| | Female = 569 (49.2%) | Female = 644 (50.3%) |
| Musculoskeletal Pain (at baseline) | Yes = 288 | Yes = 323 |
| | No = 868 | No = 958 |
| Work Satisfaction | Satisfied = 1045 | Satisfied = 1167 |
| | Dissatisfied = 111 | Dissatisfied = 114 |

In unadjusted models, wealth, marital status, and physical job demands did not influence the time to retirement or reporting being out of work and were therefore not included in subsequent analyses. Next, we used the remaining significant univariate predictors to run two multivariate Cox proportional hazard regression analyses to examine how they jointly impacted on time to retirement and reporting work cessation in the previous month.

## Retirement

In total of 1156 not yet retired participants were included in the analysis, of which 1073 retired over the course of 14 years (Fig 2). Work dissatisfaction was associated with earlier retirement compared to reporting satisfaction with work (HR = 1.29, CI = 1.03–1.62). Those reporting musculoskeletal pain complaints tended to retire earlier compared to pain free participants (HR = 1.30, CI = 1.12–1.49). Female participants had a 1.27-increased risk (CI = 1.13–1.44) of retiring earlier compared to male participants. Higher self-perceived social status was associated with earlier retirement age (HR = 1.01, CI = 1.00–1.01). Participants who feel they receive the recognition they deserve at work tended to retire at a later age compared to those who disagreed with this statement (HR = 0.78, CI = 0.66–0.91). Older age at baseline was also associated with retiring later (HR = 0.89, CI = 0.88–0.91), but this is likely owed to the statistical approach of using years, and by extension age, to indicate survival time, i.e., age at retirement. Those having a higher age at baseline tended to retire at a later age, because they were already older at that point and everyone was followed for the same duration, i.e., 14 years, and so this effect merely suggests that older people who were still working retired at an older age. Severity of depressive symptoms, psychosocial job demands, decision authority, and social support did not influence how soon participants retired in the adjusted model.

Repeating the analysis, but excluding participants who retired within the two years after baseline (sensitivity analysis), job satisfaction no longer had a significant impact. None of the other associations between the potential determinants and retirement were affected.

## Work cessation

A similar approach was employed to examine the contribution of the predictor variables on work cessation. This analysis comprised of 1281 (mean age 58.3, *SD* = 4.5), participants who

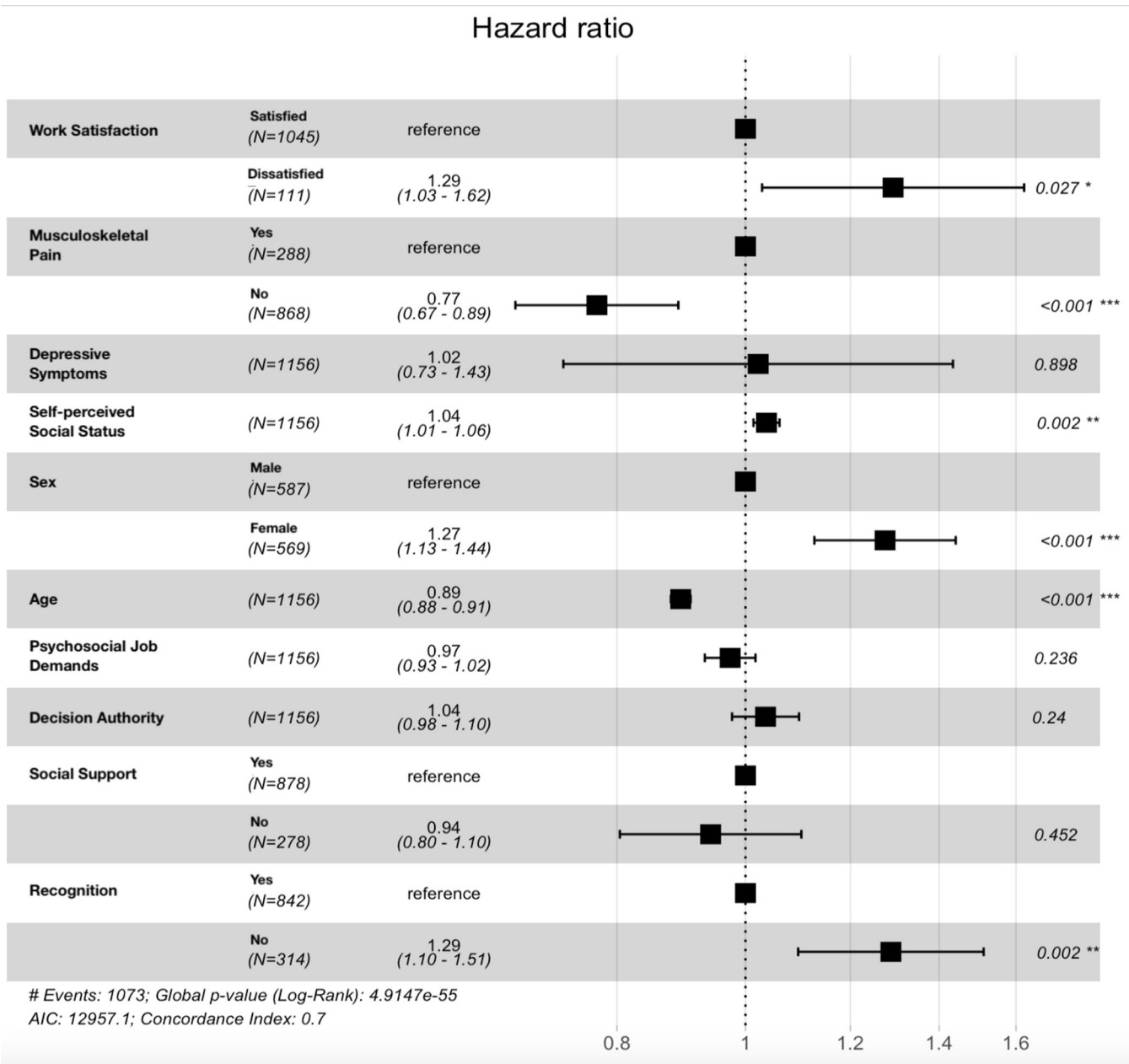

**Fig 2. Hazard ratios for retirement.** ***values are rounded. Please note that for self-perceived social status the hazard ratio indicates the change in risk per five-point increment on a scale from 0–100.

were in employment at wave 2 and for whom complete data were available. Over the 14-year period participants were followed, 1196 reported work cessation (Fig 3). Work dissatisfaction was associated with an increased risk of ceasing work at an earlier age (HR = 1.30, CI = 1.04–1.62). Participants suffering from musculoskeletal pain were 1.25 times more likely to cease work sooner (CI = 1.10–1.43). Higher perceived social status was associated with earlier reports of work cessation (HR = 1.01, CI = 1.0–1.01). Female participants were more likely to report work cessation at an earlier age compared to male participants (HR = 1.28, CI = 1.14–1.43). Like the analysis above, age was spuriously associated with being of older age when

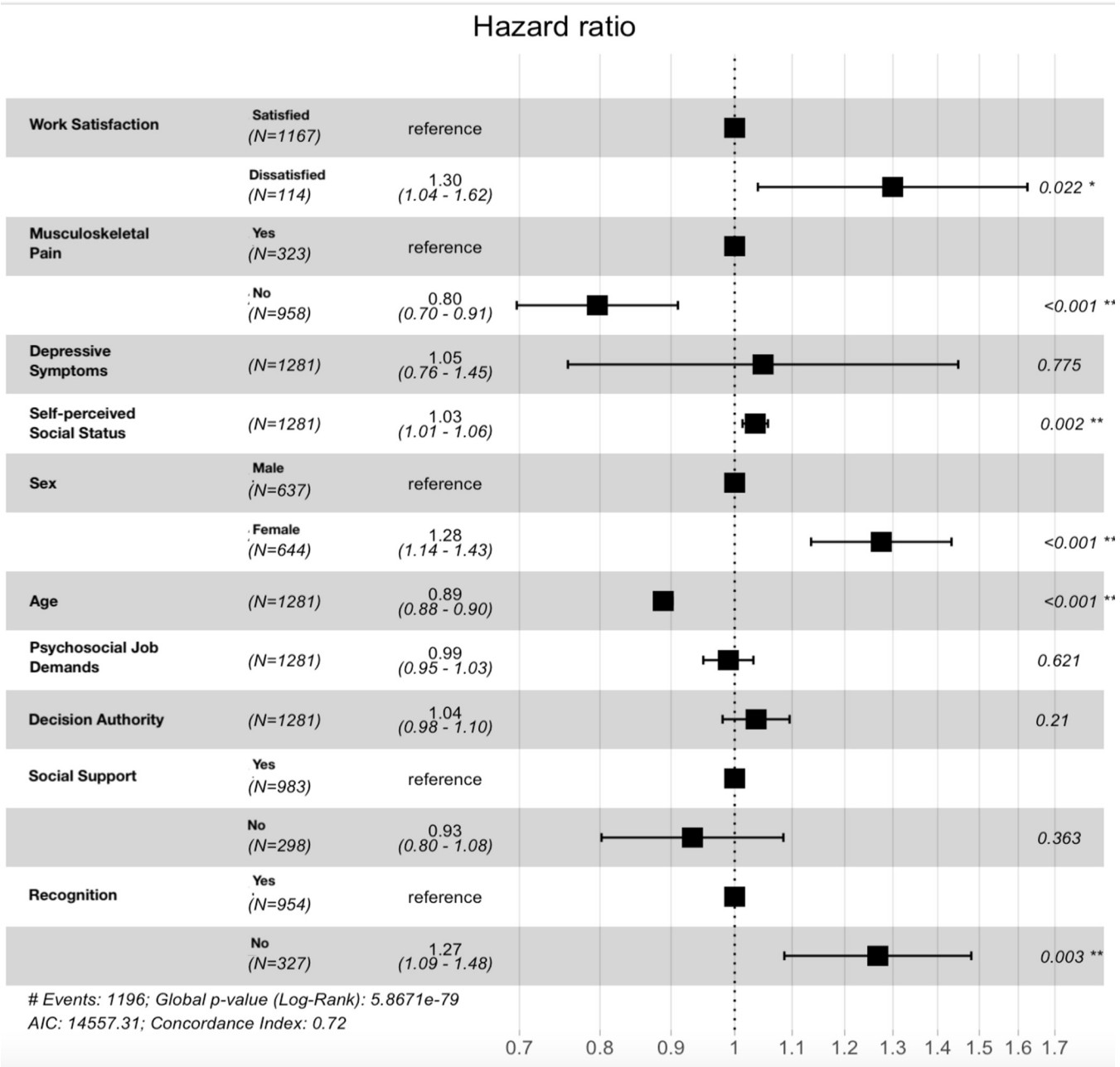

**Fig 3. Hazard ratios for Wok Cessation.** ***values are rounded. Please note that for self-perceived social status the hazard ratio indicates the change in risk per five-point increment on a scale from 0–100.

reporting work cessation (HR = 0.89, CI = 0.88–0.90). Finally, not receiving the recognition they felt they deserved in their job meant that those participants tended to report work cessation at a younger age (HR = 1.27, CI = 1.09–1.48). Severity of depressive symptoms, psychosocial job demands, decision authority, and social support did not influence the age at which participants reported work cessation.

Repeating the analysis, but excluding participants who reported work cessation within the two years following baseline (sensitivity analysis), job satisfaction no longer had a significant

impact. None of the other associations between the potential determinants and work cessation were affected.

## Discussion

Our analyses have demonstrated that frequent musculoskeletal (MSK) pain was significantly associated with the risk of earlier retirement and of work cessation in a sample of older workers based in England. Frequent musculoskeletal pain remained a significant predictor of earlier retirement and risk of work cessation at earlier ages when controlling for the influence of job satisfaction, depressive symptoms, self-perceived social status, sex, and working conditions.

As well as MSK pain, several predictor variables significantly associated with increased risk of retiring or leaving the workforce earlier. It is perhaps unsurprising that being female associated with leaving the workforce earlier, given that the UK state pension age was lower for females than males during the study period, being equalised in 2018. Mandatory retirement has been abolished in the UK since 2011 which, at least notionally, means workers can choose when to retire [21]. This gives greater scope for work-related factors to play a role in decisions to leave the workforce. As such job satisfaction, a perceived combination of many work-related factors, was a significant predictor in both our analyses. This is in line with a Danish study in general workers where lower work satisfaction has been found to associate with increased risk of retirement [30]. However other studies have demonstrated inconsistent effects of satisfaction in general worker retirement [31], which may partially explain why satisfaction was not a significant predictor in the sensitivity analysis. Finally, recognition or appreciation at work has been associated with retirement in several cohorts of general workers [28,30,32]. Our study reflects these findings in a model that includes chronic MSK pain.

Although we have presented the results of separate analyses of retirement and leaving employment, it should be noted that there is considerable crossover in the analytical samples. For example, a participant with frequent MSK pain who retired early, will also have left employment early. However, the outcomes are conceptually different which justified the dual analysis approach. As with any self-reported predictors there is a chance of recall bias. However, ELSA is a longitudinal data set which minimises such bias as the predictors were measured before the outcomes, which may have occurred many years later. The sample may also be biased in that people with serious MSK pain may have already left the workforce prior to baseline, therefore our results may be an underestimate of the effect of MSK pain on workforce exit. The longitudinal nature of the analysis does not allow for changes in a participant's working environment or changes in their perception of the same. Similarly, it does not account for changes in the presence of musculoskeletal pain complaints or their intensity. For example, physical job demands (excluded from our analysis at the unadjusted model stage) may not pose a problem to participants when measured at baseline but may become troublesome as the person ages and approaches state retirement age [23]. Further due to the binary nature of the MSK pain question, it was not possible to analyse if different levels of severity of MSK pain have a relationship with the outcomes. Finally, it is important to note that our study was undertaken in an English cohort, and it is not clear if these results will apply to other work and social security contexts.

It is well established that poor health can increase risk of retirement and unemployment [33,34]. However, Fisher et al [35] notes that the relationship is not linear as good health can also encourage earlier retirement, especially amongst those who are in a financial position to do so. Therefore, it is notable that in our study frequent MSK pain associated with both earlier work cessation and retirement.

More specifically this study complements results from a recent Danish study of older workers with physically demanding work where work limiting pain increased the risk of loss of employment [13]. Our study demonstrated that frequent MSK pain associated with poor work outcomes in a sample of general workers. Our study also utilises a wider definition of pain, which is notionally independent of, but may include, functional interference from pain or work limiting pain which has also been shown to lead to poor work outcomes [13,36]. In a large cohort of Swedish workers sickness absence due to lower back pain was associated with increased disability pensions and early retirement [37]. Our results are largely in line with this, although our measure of MSK pain is again wider as it did not limit the analysis to sickness absence attributed to pain.

Our study does not identify the reasons that people with frequent MSK pain may have left the workforce. De Wind et al [38], identified that poor health can push Dutch workers towards retirement for several reasons, including being unable to work or concern that health may decline in the future. It is therefore possible that participants in our study left work for several reasons including their pain making it impossible to continue, or a perception that work may increase their pain in the future. It is also unclear whether the people with MSK pain were pushed out by employers or have agency in deciding to leave the workforce, for example because of lowered productivity [18] or a declining sense of contributing to the workplace [39]. Therefore, identifying the reasons that people with pain may leave the workforce would be vital to understand what can be done to extend working lives for people with pain.

This study adds a longitudinal analysis of the relationship between frequent MSK pain and employment and retirement in an English sample. Our measure of pain is wider than functionally limiting pain and may suggest that a wider range of pain experiences can also lead to poor work outcomes.

## Conclusion

Frequent MSK pain may increase the risk of earlier work cessation and earlier retirement. Further research should establish the mechanisms and decision making involved in leaving the workforce in people with frequent MSK pain.

## Author Contributions

**Conceptualization:** Nils Georg Niederstrasser, Elaine Wainwright, Martin J. Stevens.

**Data curation:** Nils Georg Niederstrasser.

**Formal analysis:** Nils Georg Niederstrasser.

**Methodology:** Elaine Wainwright, Martin J. Stevens.

**Writing – original draft:** Nils Georg Niederstrasser, Elaine Wainwright, Martin J. Stevens.

**Writing – review & editing:** Nils Georg Niederstrasser, Elaine Wainwright, Martin J. Stevens.

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
