## [Decision Letter · Decision Letter 0]

30 Oct 2023

PONE-D-23-31458Pain affects the age of retirement and the risk of work cessation among older peoplePLOS ONE

Dear Dr. Niederstrasser,

Thank you for submitting your manuscript to PLOS ONE. After careful consideration, we feel that it has merit but does not fully meet PLOS ONE’s publication criteria as it currently stands. Therefore, we invite you to submit a revised version of the manuscript that addresses the points raised during the review process.

ACADEMIC EDITOR: Thank you very much for submitting your manuscript to PLOS ONE. In addition to the concerns raised by the reviewers, please reorganize the methods section in accordance with the STROBE guidelines. The current format makes it challenging to follow the methods.

We look forward to receiving your revised manuscript.

Kind regards,

Amin Nakhostin-Ansari

Academic Editor

PLOS ONE

Journal Requirements:

Reviewers' comments:

Reviewer's Responses to Questions

**Comments to the Author**

1. Is the manuscript technically sound, and do the data support the conclusions?

Reviewer #1: Yes

Reviewer #2: Yes

2. Has the statistical analysis been performed appropriately and rigorously? 

Reviewer #1: Yes

Reviewer #2: Yes

3. Have the authors made all data underlying the findings in their manuscript fully available?

Reviewer #1: No

Reviewer #2: Yes

4. Is the manuscript presented in an intelligible fashion and written in standard English?

Reviewer #1: Yes

Reviewer #2: Yes

5. Review Comments to the Author

Reviewer #1: The data is valuable given the aging of populations around the world.

Therefore, it is advisable to provide detailed information about the subjects of this study. In other words, what kind of work where they engaged in, what was the age distribution of their subjects, and what was the ratio of men to women? It also raises questions about whether they haven't been able to gather information about the location and duration of pain. Therefore, it is desirable to identify what information you can or cannot collect about the subjects of this study. In addition, data that can be presented should be presented in a table.

Interestingly, musculoskeletal pain has been shown to be an independent predictor of early retirement and job interruption. However, information such as pain intensity, location, duration, and presence or absence of disease may be important for planning measures, but can you provide information on these points? Are these data not necessary to achieve the objectives of this study?

The study found that frequent musculoskeletal pain remained a significant predictor of early retirement and the risk of leaving early, even after controlling for the effects of job satisfaction, depressive symptoms, self-perceived social status, gender, wealth, and working conditions. Your argument against this result is lacking.

Are these results considered reasonable? Will the results depend on the cohort in this study, on factors such as English language differentiation, or will they be seen as common results globally? I think you can refer to it from multiple angles.

Reviewer #2: The topic of this manuscript, dealing with consequences of musculoskeletal pain among individuals 50 years and more, is interesting. The data are issued from the English Longitudinal Study of Aging, which includes information about work cessation and retirement, with a rather long follow-up. The subjects are classified (at baseline) as often troubled by bone, joint or muscle pain, yes or no (self-report).

The manuscript should be improved in order to be easier to understand for the reader:

1 - The title could be modified: « musculoskeletal pain » rather than « pain »

2 - The study design must be explained more precisely: data from wave 2 (baseline) were used for the predictors. Retirement and work cessation were recorded from wave 2 to wave 9, fourteen years later. For retirement, 1156 workers at baseline, 1073 retired over the course of 14 years. For work cessation, the corresponding numbers are 1281 and 1197. As if no one had been lost to follow-up? It is said that additional participants were added with each wave of the cohort to maintain the total number of participants. Are there “additional participants” included in the present study?

3 - The potential effect of legal age at retirement is discussed, but it would be useful to have some information about the context of retirement in England earlier in the paper, especially the fact that “workers can choose when to retire”.

4 – Work cessation and retirement: many subjects are in both categories. This is said in the discussion, but it should appear earlier in the paper, with numbers (how many subjects belonging to both categories). Would it be possible to study the category: work cessation (no paid employment last month) but not retired?

5 - Participants’ age was used to denote survival time in the Cox models. This is probably not very usual. It seems that it explains the results observed for age. Does it explain that physical job demands seems not to be associated with time to retirement? it is also surprising that a higher self-perceived social status is associated with an increased risk of ceasing work and retiring at an earlier age.

6 – Marital status is indicated as a variable in “results” but not included in the list of predictors.

7 - For retirement, HR=0.77 for report of musculoskeletal complains. This HR is probably for « no report », one expects an HR larger than 1, associated with report of pain (as for work cessation).

8 – In figures 1 and 2, since some variables are dichotomous, and other ones quantitative, it is not obvious to compare the HRs. It would be useful to have a footnote indicating, for some variables, that the HR corresponds to a change of one unit for the variable. The range of the scores should be clearly presented in the text, and added in table 1: 0 (?) to 8 for depression and psychosocial demands (but the mean is 0.1 for depressive symptoms in table 1), 1 to 100 for social status (leading to a significant HR equal to 1.01!). For decision authority, there is no indication about the range in the text.

Minor comments

Abstract, first line of « result »: « was » is lacking after « pain ». Same in the first line of discussion.

« Frequent musculoskeletal pain » in the list of predictors: delete « they responded ».

Figures 1 and 2: “depression” in figure 1, “depressive symptoms” in figure 2. Probably the same variable.

(second) reference Bevan: some information is lacking (journal…)

6. PLOS authors have the option to publish the peer review history of their article (what does this mean?). If published, this will include your full peer review and any attached files.

Reviewer #1: No

Reviewer #2: No

---

## [Author Response · Author response to Decision Letter 0]

9 Nov 2023

Academic Editor: Please reorganize the methods section in accordance with the STROBE guidelines. The current format makes it challenging to follow the methods.

We have rearranged the methods section following the strobe guidelines.

Reviewer #1: 

Thank you for your suggestions. We have incorporated them and feel this has greatly improved the manuscript.

The data is valuable given the aging of populations around the world. Therefore, it is advisable to provide detailed information about the subjects of this study. In other words, what kind of work where they engaged in, what was the age distribution of their subjects, and what was the ratio of men to women? It also raises questions about whether they haven't been able to gather information about the location and duration of pain. Therefore, it is desirable to identify what information you can or cannot collect about the subjects of this study. In addition, data that can be presented should be presented in a table.

Thank you for the suggestions. We have added more information pertaining to the sample to Table 1, including the age distribution and ratio of men to women. 

Interestingly, musculoskeletal pain has been shown to be an independent predictor of early retirement and job interruption. However, information such as pain intensity, location, duration, and presence or absence of disease may be important for planning measures, but can you provide information on these points? Are these data not necessary to achieve the objectives of this study?

Some information is available pertaining to pain intensity and type of work; however, this is difficult to incorporate, as the data span 14 years and the resolution of the available information is low, as assessments took place every two years. Therefore, the value of providing information pertaining to pain intensity and type of work for each participant may be misleading as there may be multiple changes throughout each interval that are not captured. The finding that mere presence of musculoskeletal pain has such a deleterious impact does, in our opinion, achieve the study’s objectives. We have clarified the discussion section to reflect this issue (see pg. 12).

The study found that frequent musculoskeletal pain remained a significant predictor of early retirement and the risk of leaving early, even after controlling for the effects of job satisfaction, depressive symptoms, self-perceived social status, gender, wealth, and working conditions. Your argument against this result is lacking.

Are these results considered reasonable? Will the results depend on the cohort in this study, on factors such as English language differentiation, or will they be seen as common results globally? I think you can refer to it from multiple angles.

You raise an interesting point. The English Longitudinal Study of Ageing is representative of individuals aged 50 and over residing in England and so generalisability to other nations may be limited. Nevertheless, although the findings likely do apply to other nations with similar demographic profiles and are likely to hold global merit our results do not go this far and so we have acknowledged this limitation in the discussion section (see pg. 13). However, we do consider these results to be reasonable, as they match similar previous investigations. 

Reviewer #2: 

We thank reviewer 2 for their comments and suggested revisions. We have taken these on board and revised the manuscript to improve clarity. We feel the paper has been greatly improved following revisions based on suggestions by reviewer 2.

1. The title could be modified: « musculoskeletal pain » rather than « pain »

Thank you for the suggestion. We have made the amendment.

2. The study design must be explained more precisely: data from wave 2 (baseline) were used for the predictors. Retirement and work cessation were recorded from wave 2 to wave 9, fourteen years later. For retirement, 1156 workers at baseline, 1073 retired over the course of 14 years. For work cessation, the corresponding numbers are 1281 and 1197. As if no one had been lost to follow-up? It is said that additional participants were added with each wave of the cohort to maintain the total number of participants. Are there “additional participants” included in the present study?

There were no additional participants added to the study during the follow up period that were included in the analysis. The mention in the paper was a reference to the general data collection and recruitment methods employed by the ELSA team. Each wave (every two years), due to the age of participants, additional participants are added from that year’s Health Survey for England (HSE) to maintain a representative sample. Our study followed participants present in ELSA at wave 2, which means at wave 2 additional participants from the HSE were added to boost the numbers and maintain a representative sample. We only used data from participants present in wave 2 onwards in our analyses. We have changed the wording in the method section to clarify this point.

The entire data set at baseline data (wave 2) comprised 8780 participants (before removal of incomplete data) out of which 2405 reported being in employment and 2582 reported not being retired and so were eligible for inclusion in the study. After removal of missing values due to incomplete data and loss to follow up the final numbers were 1281 and 1156 eligible participants respectively. We have amended the wording in the method section to clarify this point (see pg. 5-6).

3. The potential effect of legal age at retirement is discussed, but it would be useful to have some information about the context of retirement in England earlier in the paper, especially the fact that “workers can choose when to retire”.

Thank you. We have added a short paragraph into the introduction with a brief overview of retirement in the UK which has been subject to several changes since 2010 (see pg. 4). 

4. Work cessation and retirement: many subjects are in both categories. This is said in the discussion, but it should appear earlier in the paper, with numbers (how many subjects belonging to both categories). Would it be possible to study the category: work cessation (no paid employment last month) but not retired?

We have added the number of participants (1156) belonging to both categories to the text (see pg. 6).

This would indeed be a fascinating and worthwhile analysis. We looked into this, and selecting those who were not in paid employment in the last month and had not retired yet led to 91 eligible participants after removing participants with missing data for any of the relevant variables. Nevertheless, the exact analysis cannot be repeated, as these participants, by definition, do not have any data pertaining to job demands and characteristics and so the results would not be comparable to the current analyses. Furthermore, the resultant low sample size invites prudence in interpreting the results from this analysis. We therefore decided that, in the current context, this additional analysis does not bear any additional merit to the paper.

5. Participants’ age was used to denote survival time in the Cox models. This is probably not very usual. It seems that it explains the results observed for age. Does it explain that physical job demands seems not to be associated with time to retirement? it is also surprising that a higher self-perceived social status is associated with an increased risk of ceasing work and retiring at an earlier age.

High physical job demands are associated with occupations involving manual labour. These jobs are further associated with lower pay which often precludes the individual from retiring at the point that the physical aspects of their job become too much to continue working. As a result, changes to less physically demanding jobs often ensue that result in continuous employment, albeit in different sectors.

Higher self-perceived social status is often associated with wealth or perceived wealth. It is therefore reasonable to assume that those with higher self-perceived social status may have higher perceived or actual wealth and so have the option of retiring earlier. In other words, they may be able to afford not to work, as they can live off savings and pensions.

6. Marital status is indicated as a variable in “results” but not included in the list of predictors.

Thank you for pointing out this oversight. This has been amended (see pg. 9).

7. For retirement, HR=0.77 for report of musculoskeletal complains. This HR is probably for « no report », one expects an HR larger than 1, associated with report of pain (as for work cessation).

Thank you for pointing this out. We realise that we were not clear in how this was reported in the manuscript. You are correct that the HR below 1 is for the reverse, i.e., “no report”, and the number in the text should in fact be the inverse, hence, 1.30. We have changed the values to clarify this (see pg. 10).

8. In figures 1 and 2, since some variables are dichotomous, and other ones quantitative, it is not obvious to compare the HRs. It would be useful to have a footnote indicating, for some variables, that the HR corresponds to a change of one unit for the variable. The range of the scores should be clearly presented in the text, and added in table 1: 0 (?) to 8 for depression and psychosocial demands (but the mean is 0.1 for depressive symptoms in table 1), 1 to 100 for social status (leading to a significant HR equal to 1.01!). For decision authority, there is no indication about the range in the text.

We thank the reviewer for this suggestion. We forgot to include that the summed score for the number of depressive symptoms was divided by 8, resulting in scores ranging from 0 to 1. Self-perceived social status is operationalised by dividing the range of 0 – 100 into 20 five-point increments. Therefore, the range is 0 – 20 and each single unit increase there, represents a five-point increase on a scale from 0 – 100. We initially presented the mean on a scale from 0 – 100, but realise that this may lead to confusion and have reverted to reporting the mean on a scale from 0 – 20 to better illustrate the impact of a single unit increase on this measure. Ranges have been added where missing in the method section (see pg. 7 – 9) and Table 1.

Footnote to be added to both Figures: Please note that for self-perceived social status the hazard ratio indicates the change in risk per five-point increment on a scale from 0 – 100.

Minor comments

Abstract, first line of « result »: « was » is lacking after « pain ». Same in the first line of discussion.

« Frequent musculoskeletal pain » in the list of predictors: delete « they responded ».

Figures 1 and 2: “depression” in figure 1, “depressive symptoms” in figure 2. Probably the same variable.

(second) reference Bevan: some information is lacking (journal…)

Thank you for pointing out these oversights, which we have corrected.

---

## [Decision Letter · Decision Letter 1]

13 Dec 2023

PONE-D-23-31458R1Musculoskeletal pain affects the age of retirement and the risk of work cessation among older peoplePLOS ONE

Dear Dr. Niederstrasser,

Thank you for submitting your manuscript to PLOS ONE. After careful consideration, we feel that it has merit but does not fully meet PLOS ONE’s publication criteria as it currently stands. Therefore, we invite you to submit a revised version of the manuscript that addresses the points raised during the review process.

We look forward to receiving your revised manuscript.

Kind regards,

Amin Nakhostin-Ansari

Academic Editor

PLOS ONE

Journal Requirements:

Reviewers' comments:

Reviewer's Responses to Questions

**Comments to the Author**

1. If the authors have adequately addressed your comments raised in a previous round of review and you feel that this manuscript is now acceptable for publication, you may indicate that here to bypass the “Comments to the Author” section, enter your conflict of interest statement in the “Confidential to Editor” section, and submit your "Accept" recommendation.

Reviewer #2: All comments have been addressed

Reviewer #3: (No Response)

2. Is the manuscript technically sound, and do the data support the conclusions?

Reviewer #2: Yes

Reviewer #3: (No Response)

3. Has the statistical analysis been performed appropriately and rigorously? 

Reviewer #2: Yes

Reviewer #3: (No Response)

4. Have the authors made all data underlying the findings in their manuscript fully available?

Reviewer #2: Yes

Reviewer #3: (No Response)

5. Is the manuscript presented in an intelligible fashion and written in standard English?

Reviewer #2: Yes

Reviewer #3: (No Response)

6. Review Comments to the Author

Reviewer #2: The comments raised in the previous version have been correctly adressed. This revised manuscript is easy to read and to understand

Reviewer #3: I had the opportunity to review this interesting manuscript on effects of MSK pain on retirement age among older people. The idea is interesting and the manuscript is well-written. There are a few areas where I believe some enhancements could further elevate the quality of the manuscript.

a. According to the title the study aimed to evaluate the effects of MSK pains on work cessation and more focus should be place on MSK pains in the introduction section rather than discussing chronic pain in general.

b. Please briefly describe the sampling and data collection process.

c. Please consider showing the flow of participant in a figure (method section – participant subheading).

d. Please move the variables subheading to the next line.

7. PLOS authors have the option to publish the peer review history of their article (what does this mean?). If published, this will include your full peer review and any attached files.

Reviewer #2: No

Reviewer #3: No

---

## [Author Response · Author response to Decision Letter 1]

18 Dec 2023

Academic Editor: Please review your reference list to ensure that it is complete and correct. If you have cited papers that have been retracted, please include the rationale for doing so in the manuscript text, or remove these references and replace them with relevant current references. Any changes to the reference list should be mentioned in the rebuttal letter that accompanies your revised manuscript. If you need to cite a retracted article, indicate the article’s retracted status in the References list and also include a citation and full reference for the retraction notice.

We have reviewed the reference list and made the following changes:

Added Waddell & Burton 2006, Taylor 2017 due to omission from reference list. 

Amended De Wind 2014 to De Wind 2013 in the manuscript (the reference list is correct to cite both).

Amended Bevan 2015 to Bevan 2015b, due to oversight.

Added Fimland et al. 2018 in introduction due to reviewer 3’s suggestion to focus more on musculoskeletal pain.

Reviewer #2: The comments raised in the previous version have been correctly addressed. This revised manuscript is easy to read and to understand.

We thank reviewer 2 for the comments and suggestions, which we believe have greatly improved the paper.

Reviewer #3: I had the opportunity to review this interesting manuscript on effects of MSK pain on retirement age among older people. The idea is interesting and the manuscript is well-written. There are a few areas where I believe some enhancements could further elevate the quality of the manuscript.

Thank you for your comments and suggestions. We have revised the manuscript accordingly. 

1. According to the title the study aimed to evaluate the effects of MSK pains on work cessation and more focus should be place on MSK pains in the introduction section rather than discussing chronic pain in general.

Thank you for bringing this up. We have made changes to the introduction to highlight the manuscript’s focus on MSK pain, see page 3.

2. Please briefly describe the sampling and data collection process.

We have added additional details pertaining to the sampling and data collection process on pages 5 and 6.

3. Please consider showing the flow of participant in a figure (method section – participant subheading).

Thank you for this suggestion. We have included a figure.

4. Please move the variables subheading to the next line.

We have moved the subheadings to the next line.

---

## [Decision Letter · Decision Letter 2]

2 Jan 2024

Musculoskeletal pain affects the age of retirement and the risk of work cessation among older people

PONE-D-23-31458R2

Dear Dr. Niederstrasser,

We’re pleased to inform you that your manuscript has been judged scientifically suitable for publication and will be formally accepted for publication once it meets all outstanding technical requirements.

Kind regards,

Amin Nakhostin-Ansari

Academic Editor

PLOS ONE

Additional Editor Comments (optional):

Reviewers' comments:

Reviewer's Responses to Questions

**Comments to the Author**

1. If the authors have adequately addressed your comments raised in a previous round of review and you feel that this manuscript is now acceptable for publication, you may indicate that here to bypass the “Comments to the Author” section, enter your conflict of interest statement in the “Confidential to Editor” section, and submit your "Accept" recommendation.

Reviewer #3: All comments have been addressed

2. Is the manuscript technically sound, and do the data support the conclusions?

Reviewer #3: Yes

3. Has the statistical analysis been performed appropriately and rigorously? 

Reviewer #3: Yes

4. Have the authors made all data underlying the findings in their manuscript fully available?

Reviewer #3: (No Response)

5. Is the manuscript presented in an intelligible fashion and written in standard English?

Reviewer #3: (No Response)

6. Review Comments to the Author

Reviewer #3: I have read the revised manuscript and I am satisfied with the changes made by the authors. They have improved the clarity, accuracy, and relevance of their paper.

7. PLOS authors have the option to publish the peer review history of their article (what does this mean?). If published, this will include your full peer review and any attached files.

Reviewer #3: No

---

## [Editor Report · Acceptance letter]

24 Feb 2024

PONE-D-23-31458R2 

PLOS ONE

Dear Dr. Niederstrasser, 

I'm pleased to inform you that your manuscript has been deemed suitable for publication in PLOS ONE. Congratulations! Your manuscript is now being handed over to our production team.

Kind regards, 

on behalf of

Dr. Amin Nakhostin-Ansari 

Academic Editor

PLOS ONE